# What Predicts Adherence to Governmental COVID-19 Measures among Danish Students?

**DOI:** 10.3390/ijerph18041822

**Published:** 2021-02-13

**Authors:** Gabriele Berg-Beckhoff, Julie Dalgaard Guldager, Pernille Tanggaard Andersen, Christiane Stock, Signe Smith Jervelund

**Affiliations:** 1Unit for Health Promotion Research, Department of Public Health, University of Southern Denmark, 6700 Esbjerg, Denmark; jguldager@health.sdu.dk (J.D.G.); ptandersen@health.sdu.dk (P.T.A.); christiane.stock@charite.de (C.S.); 2Unit for Health Research, University of Southern Denmark, Hospital South West Jutland, 5000 Odense, Denmark; 3Research Department, University College South Denmark, 6100 Haderslev, Denmark; 4Institute of Health and Nursing Science, Berlin Institute of Health, Freie Universität Berlin, Charité—Universitätsmedizin Berlin, 13353 Berlin, Germany; 5Department of Public Health, Section for Health Services Research, University of Copenhagen, 1014 Copenhagen, Denmark; ssj@sund.ku.dk

**Keywords:** COVID-19, risk behavior, students, governmental recommendation predictors

## Abstract

Knowledge on compliance with governmental recommendations in combating the spread of COVID-19 in different groups is important to target efforts. This study investigated the adherence to the governmental implemented COVID-19 measures and its predictors in Danish university students, a not-at-risk group for COVID-19 mortality and normally characterized by many social contacts. As part of the COVID-19 International Student Wellbeing Study, a survey on socio-demographic situation, study information, living arrangements, lifestyle behaviors, stress, questions about COVID-19 infection and knowledge and concern about COVID-19 infection was sent via email to relevant university students in Denmark in May, 2020 (*n* = 2.945). Stepwise multiple linear regression analysis was employed. Our results showed that around 60% of the students were not concerned about COVID-19, while 68% reported that they followed governmental measures. The main facilitators for following the recommendations were older age, concern about COVID-19 and depression, while barriers were living in a student hall, being physical active or reporting mental stress. Only 9% of the variation in adhering to governmental recommendations could be explained by the analyzed predictors. Results may inform health communication. Emotionally appealing information rather than knowledge-based information may be more effective in motivating students to follow COVID-19 measures.

## 1. Introduction

Compliance with COVID-19 governmental recommendations is important to combat the spread of this pandemic disease. In order to develop target group-specific health information and to communicate it effectively, it is important to know what types of people adhere to the recommendations and what types of people do not. Several studies have investigated predictors in precautionary behavior to protect against COVID-19 infection [1,2,3,4]. A Japanese survey showed that participants who were reluctant to implement proper prevention measures were more likely to be male, younger, unmarried, with lower income household and a smoker or drinker [1]. In a Canadian survey among children, screen time, outdoor activities and restrictive capacities from parents predicted adherence to recommendations against COVID-19 infections [2]. Data on a large international survey in North America and Europe on the adherence of social distancing recommendations mentioned as facilitators self-protection, feeling responsible to protect the community and being able to work. Barriers were having friends who need help and to avoid feeling lonely [3]. In an analysis of two Norwegian cohorts, compliance with governmental recommendations was lower than expected in a country with a high trust in the government. Compliance was largely unaffected by age and lower in men and women with the highest educational level [4]. However, knowledge about predictors of adherence is scarce in Denmark.

Governmental recommendations in combating the spread of COVID-19 require citizens to make changes in their behavior. Factors that influence compliance mentioned in the literature are trust and confidence in the government, knowledge about the risk, social experience, mental health and wellbeing [5]. Trust in the government can assure citizens that guidelines are necessary and effective [4]. The greater the knowledge about the infection, the more it is expected that individuals will follow the recommendation. This is explained by the health belief model positing that health behavior decisions are determined by weighting the costs and benefits of different health-related actions [6]. Social experience means that within a group, specific informal rules are shared, which affects the behavior of individuals and the group in following governmental recommendations [7]. Finally, the association between compliance and mental health and wellbeing is still conflicting [8]. Furthermore, we suggest that the overall health behavior and preexisting chronic disease might facilitators in following governmental recommendations.

Denmark has reacted to the COVID-19 pandemic with prompt shutdown but also a fast reopening. On February the 27th 2020, the first case of COVID-19 was confirmed. On 12 March 2020, a rapid increase in the numbers of infected people made the Danish Government start comprehensive political initiatives and shut down the country [9]. The basic elements of the strategy were requirements and recommendations for physical distance, social isolation, assembly bans, ban for visiting vulnerable family members and friends and the use of protective equipment in public spaces, especially regarding hand hygiene [10]. Regarding personal protective measures, the recommendations focused on hand hygiene and respiratory etiquette without any recommendations for the population to use face masks. Governmental bans included shutdowns of daycare, all educational institutions and sports facilities, shopping malls, shops and businesses, restaurants, cafes, clubs and pubs, as well as the closure of borders for entry, except for people living or working in Denmark, Danish citizens or visitors with a “relevant purpose” [9]. Universities were closed, and teaching was offered on-line. All classes and exams were rescheduled immediately into online teaching.

Reopening started on 15 April 2020. Daycare and specific younger classes in schools were prioritized to reopen first [11]. From mid-May, all students in compulsory and upper secondary school returned to school, but most of the university education was kept online until August 1, 2020. In consideration of the economic consequences that the shutdown has caused, private companies were prioritized in the first and second phases of the reopening, almost returning to normal conditions in May. Parts of the public sector and universities still had to work from home until mid-June. However, this re-opening was marked by regional differences. The eastern part of Denmark had prolonged workplace restrictions, as the number of infections remained high. Cultural activities, indoor sports activities and night clubs were among the areas which were re-opened step by step from May onward. The travel restriction was loosened in end-May starting with business trips as a valid purpose, as the Danish economy is strongly dependent on export. Applicable purposes were added to the list for people travelling from Germany and the Nordic countries if owning a summerhouse, having a relationship, visiting family or having a job interview [12]. More information is available via the WHO-COVID-19 health system monitor [9]. In the beginning of the pandemic, the overall population supported governmental activities. The reopening phase was characterized by greater political disagreement, regional differences and unwillingness to follow the recommendations [13]. Our survey study took place during this reopening phase, in which the support for the COVID-19 measures started to erode in society.

University students’ lives are among the areas that are most affected by the COVID-19 pandemic. Online teaching was most prolonged in this group compared to other educational settings; electronic exams were introduced for all types of exam; job options were reduced dramatically; and potential international exchange plans were stopped immediately. On top of these dramatic changes, the private and social lives of students were severely changed due to lacking sport or other leisure activities, and restrictions to meet together challenged social bonds. Furthermore, youthhood is often characterized by many social contacts and as a transition between child and adult where you develop a deeper awareness of personal responsibility and interdependence as members of society [14]. Young people are not in a risk group for COVID-19 mortality as such [15] but are still able to spread the virus to COVID-19 vulnerable groups.

Therefore, we aimed to analyze the adherence to the COVID-19 measures implemented by the government and its predictors in Danish university students. Specifically, we explored if sociodemographic variables, knowledge about COVID-19, mental health and concern about the disease were associated with the self-reported adherence to governmental recommendations during the reopening phase in May and June 2020.

## 2. Method

The study is part of the COVID-19 International Student Wellbeing Study [16]. Survey participation was voluntary and anonymous, and data were confidential and protected. The study adheres to Danish standards for ethical conduct of scientific studies and was approved by the Research Ethics Committee of the University of Southern Denmark on May 7th, 2020 (Case nr. 20/29519), and the independent ethic committee for Social Science and Humanities from the University of Antwerp, 2020 (Case: SHW_20_38).

An invitation email with the research aims and objectives was sent to all bachelor’s, master’s and PhD students at the Faculty of Health Sciences at the University of Southern Denmark on May 11th, 2020. A reminder was sent to all one week later. On June 5th, from 5394 invited students, 958 had answered (response rate of 17.8%). At the University of Copenhagen, all bachelor’s and master’s students at the Faculty of Health Science were contacted on May 29th, 2020, and no reminder was sent. In all, 766 of 7500 contacted students answered (response rate 10.2%). Students in Computer Science, Biology, Economics, Theology, Ethnology, Archaeology, Greek, Latin and History at the University of Copenhagen (app. *n* = 4900) further received the survey invitation, and notifications on specific Facebook pages generated an additional 1221 entries. In total, 2945 students participated in the study. A flow chart of data collection and analysis is presented in Figure 1.

The questionnaire collected general self-reported data on socio-demographic situation, study information, living arrangements, lifestyle behaviors and stress, questions about COVID-19 infection and students’ knowledge and concern about COVID-19 infection [16]. The survey questionnaire was also provided in English to ensure that international students could answer as well (See Appendix A).

Self-reported adherence to COVID-19 was measured with the following question: “To what degree do you adhere to the COVID-19 measures that are currently implemented by the Government?” Responses were coded on a ten-point scale from “absolutely not” to “very strictly”. For descriptive analysis, the cut-off point for students that follow governmental COVID-19 recommendations was set at 8 and above.

As socio-demographic predictors, we used sex (men, women and other), age groups (<21, 22–24, 25–30 and >30), living alone, born in Denmark, current education (bachelor’s, master’s and PhD), satisfaction with income and living situation (together with parent, student hall accommodation, with others, accommodation alone and other). Additionally, we studied Copenhagen as a place of study versus all other places of study, as most COVID-19 infections happened in the capital area.

Concern about being infected by COVID-19 was measured by the following question: “How worried are you to get infected by COVID-19?” Responses were coded in a ten-point scale from “totally not” to “very high”. Responses were recoded into 0 to 3 points: not concerned; 4 to 7 points: partly concerned; 8 and above: highly concerned. Students who reported having had the infection were treated as an extra category.

Knowledge about the infection was estimated with eight statements, such as “The virus survives for days outside the body in open air” and “You can have the virus without any symptoms”, where students could mark: “correct”, “wrong” or “I don’t know”. Correct statements were analyzed; additionally, the number of “I don’t know” statements was considered. Students were categorized as having a good knowledge, if they answered at least 7 questions correctly, and they were coded as uncertain if one or more questions was coded with “I don’t know”.

The personal connection to an infected person was estimated by the following question: “Do you know anyone in your personal network that was or currently is infected with COVID-19?”.

Chronic disease was assessed with the following question: “Do you have any of the following underlying conditions?”: cancer, diabetes, heart disease, high blood pressure, immunosuppressed conditions, kidney disease, long disease, obesity, none, prefer not to say. We chose to code none as not having a disease, and all others were coded as having any form of chronic disease.

To capture feelings of depression, we used the eight-item version of the Center for Epidemiological Studies Depression Scale (CES-D scale) [17]. It was used to indicate how much of the time during the past week students felt: (1) depressed, (2) that everything they did was an effort, (3) their sleep was restless, (4) happy, (5) lonely, (6) enjoyed life, (7) sad, (8) could not get going. A four-point Likert scale from 0 “almost none of the time” to 3 “almost all of the time” was used for each question (questions (4) and (6) were reversed) [18]. A higher score indicates the presence of more depressive symptoms. Cronbach’s Alpha was 0.85.

Cohen’s Perceived Stress Scale (PSS) in its four-item short form [19] assessed the degree to which situations in the students’ lives over the past month were appraised as stressful. (5-point scale: 0 = “never”, 1 = “almost never”, 2 = “sometimes”, 3 = “fairly often”, 4 = “very often”.) The responses were summed up so that higher scores indicated more perceived stress. Cronbach’s Alpha was 0.80.

For health behavior, vigorous physical activity was measured with the following question: “On average, how often did you perform vigorous physical activities like lifting heavy things, running, aerobics, or fast cycling for at least 30 min?” More than once a week was coded as active. Smoking was assessed with the following question: “On average, how often did you smoke tobacco (cigarettes, cigars, or e-cigarettes)?” and coded as binary: any smoking habits versus no smoking habits. Alcohol drinking was based on the question “On average, how many glasses of alcohol did you drink in one week?” and was coded as (1) never, (2) one to seven times a week and (3) more than seven times a week.

Statistical analysis was conducted in STATA 9.4 (*p* < 0.05). Stepwise multiple linear regression analysis was employed to drop irrelevant variables from the model. Model assumptions were considered graphically. To reach a better model, it was necessary to square transform the outcome. A sensitivity analysis was performed considering the sub-population of students from the Faculty of Health Sciences at the University of Southern Denmark, where the students received an e-mail and a reminder, and the response rate was the highest. This procedure was used to test if selection bias was present.

## 3. Results

The socio-demographic characteristics of the study population and the students reporting adherence to COVID-19 measures implemented by the government are presented in Table 1. Most of the students were women (78%), aged between 22 to 30 years (75%), and satisfied with their income (92%). Thirteen percent of the students were born outside Denmark. The numbers of students in bachelor’s and master’s programs were similar, while 7% of the students were PhD students. About half of the students studied in Copenhagen. There were only minor socio-demographic differences between the overall study population and the students adhering to COVID-19 measures implemented by the government.

Table 2 presents students’ concern about COVID-19, health behavior and health situation in the overall study population as well in the subgroup of students who reported adherence to COVID-19 measures. The majority of students were not concerned regarding being infected by COVID-19 (66%). At least 7 out of 8 knowledge questions were answered correctly by 60% of the students, and about 50% of students knew a person with COVID-19. Ten percent reported having chronic disease. Vigorous physical activity more than once a week was reported by 59% of the students, 89% reported not to smoke and no alcohol consumption was present in 22% of the students. Students reporting to follow the governmental recommendations were more concerned about the infection, were less knowledgeable of people in their social sphere with COVID-19, were less physically active and reported to drink less alcohol. The summary of predictors for following governmental recommendations are additionally presented in Figure 2.

The simple and stepwise multiple linear regression on all potential predictors for the adherence to COVID-19 measures is shown in Table 3. Positive predictors for following the governmental recommendations in the multiple stepwise regression model were older age, being a PhD student, having concern about the infection and feeling depressed. Barriers or negative predictors were living in a student hall, being stressed and drinking more than seven units of alcohol per week. Being single, born in Denmark, income satisfaction, place of study area, knowledge about COVID-19, having a chronic disease as well as smoking did not explain the level of adherence to governmental recommendations. However, only 9% of the variation in following governmental recommendations could be explained by the analyzed predictors.

Finally, Table 4 shows the results of the sensitivity analysis considering the more homogeneous subpopulation (one faculty of one university with the highest response rate). The results were similar to the overall results presented in Table 3, despite the fact that some variables lost significance due to a smaller sample size. One differing result was that students with a chronic disease less often adhered to COVID-19 measures in the sub-group.

## 4. Discussion

This cross-sectional survey among Danish university students describes predictors of self-reported adherence to COVID-19 measures implemented by the government. In the multiple stepwise regression model, older age, concern about COVID-19 and depression were positively associated with following governmental recommendations. In addition, living in a student hall, being physically active or having mental stress were negatively associated with following governmental recommendation. These predictors are in line with results in other studies [1,2,3,4]. However, we need to consider that only 9% of the variation in following governmental recommendations could be explained by the analyzed predictors, and many other factors remain unobserved. Overall, a high percentage (68%) reported that they followed measures implemented by the government. The high adherence can be explained by the high governmental trust given in Scandinavian countries [4]. This trust is discussed as the most important driver for following governmental recommendations [5].

One important result of our analysis is that knowledge about COVID-19 was not associated with adhering to measures implemented by the government. This result contrasts with the literature about the association of knowledge and COVID-19 protective behavior [20] stating that health knowledge supports following protective behavioral recommendations. In the study, purchasing more goods, attending large gatherings and wearing masks were considered, and the authors found different results in each health behavior and a pronounced effect of knowledge on wearing masks [20]. However, we need to consider that following governmental recommendations cannot directly be compared with single protective behavioral recommendations for health protection with a higher level of individual choice. It might be reasonable that people tend to follow governmental measures in order to ease their own decision making in an information-based society. Following this argumentation, straightforward governmental recommendations, guidelines and permission would support all citizens, independent of their level of health literacy, in their decision to follow recommendations. Furthermore, Al-Hasan et al. and Bellato [7,21] pointed out that decisions with respect to health protection are often based on emotions and not on knowledge. This argument could be supported by our findings; the concern about COVID-19 and personal contact with an infected person was more strongly associated with the self-reported adherence to governmental recommendations than knowledge about the topic. Communication strategies need to consider these insights. For adherence, it might be less important to increase knowledge about the infection, but it might be more effective to elicit positive or supportive emotions connected with following recommendations (e.g., to protect loved ones from suffering). However, one needs to keep in mind that the present results are based on students´ reports, and most respondents were affiliated with a health faculty, indicating that their level of knowledge about COVID-19 might be good. These results might be different in the general population where knowledge about health-related factors is expected to be lower. Further research is necessary.

Another interesting finding was the fact that depression was positively, and stress negatively, associated with the adherence score in the multiple regression model adjusted for all covariates. However, in the crude or bivariate model, depression and mental stress were not associated with adherence to COVID-19 measures. Similar conflicting results can be found, although results on this topic are scarce [8]. In a cross-sectional survey in 248 Australian adolescents, it could be shown that the stay-at-home adherence was not associated with depression [22]. A survey in the US with 1021 participants from the general population found that adherence to national guidelines was not associated with depression; however, adherence was negatively associated with stress [23]. Finally, a survey implemented in a behavioral weight loss intervention study with 250 mainly female participants supported a negative trend for stress (measured as post-traumatic stress disorder) and adherence to behavioral recommendations. With regard to depression, it could be shown that particular participants with moderate depression had the lowest adherence. Furthermore, estimates of effect increased when the model was adjusted for confounders such as age, sex, ethnicity, income and education [24]. However, the direction of such an association is still unclear. It is still unsolved whether better adherence leads to a lower level of stress or stress decreases adherence. Further research is warranted to clarify underlying causal mechanisms and to shed light onto the different associations in connection with adherence, stress and depression.

Having a healthy lifestyle was not consistently associated with following governmental recommendations about COVID-19 measures. For instance, being physically activity was a barrier for following governmental recommendation, and drinking alcohol was a similar barrier. It is necessary to understand the underlying reasons to understand these effects. A potential explanation is that these health behaviors are mostly carried out together with peers (such as using sport clubs or drinking a beer in a pub) and are, therefore, severely affected by following governmental recommendations to maintain social distance. For health communication purposes, this knowledge could be used for concentrating promoting messages and reminders at places where people meet (e.g., parks and sport facilities) for such activities.

A surprising result in the sensitivity analysis was that students with chronic disease were less likely to adhere to COVID-19 measurers implemented by the government. However, chronic disease was not a predictor in the overall analysis. A potential explanation for this negative and nonexistent association might be the young age and existing knowledge in the health faculty students. Very early on, it was documented that younger people do not have a risk to develop a severe disease or die when infected with COVID-19 [25].

### Strengths and Limitations

The main limitation of this study is the cross-sectional design of the study, which does not allow discussion about causality. This subjective assessment is limited due to cognitive and motivational biases, which may affect the validity of the results [26,27]. Further, the low response rate of 18 or 10% may suggest response rate bias, which might affect the estimated prevalence of participants. Those who chose to participate may be more worried about the COVID-19 outbreak and have different health behavior than those who did not participate. Therefore, prevalence should be considered with caution. However, these response rates are common in online surveys, and the sensitivity analysis supported that this selection bias is minor and did not distort the presented results. Finally, we cannot rule out that most college students had to adhere to the governmental policies, as the universities were closed, and it was impossible to enter the buildings. However, other recommendations, such as social distance, hand hygiene and meeting restrictions required their personal decisions. Furthermore, the survey took place in the reopening phase, in which the support for the COVID-19 measures started to erode in society. The controversial public discussion facilitated a free opinion formation of the students.

Strengths of this study are that the analysis is based on a very large sample, which can provide more accurate mean values and a smaller margin of error. Furthermore, the time period of data collection for this study was good. Our survey took place during this reopening phase, in which the support for the COVID-19 measures started to erode in society, which made it easier for the students to develop and present their own opinion. Finally, this survey is part of an international consortium, which allows results to be compared to those of other countries as well.

## 5. Conclusions

In conclusion, in our survey, around 60% of the students were not concerned about COVID-19. Furthermore, a high percentage (68%) reported that they followed measures implemented by the government. The main facilitators for following the recommendations were older age, concern about COVID-19 and depression, while barriers were living in a student hall, being physical active or reporting mental stress. More research is needed to explain more in depth which other factors contribute to explaining why people follow measures implemented by the government.

Finally, our results can be used for health communication. It is important to contact people at places characterized by social gatherings and where it may be necessary to change attitudes, e.g., in bars, fitness centers, sport clubs or student halls. Emotionally appealing information rather than knowledge-based information may be more effective in motivating students to follow governmental COVID-19 measures.

## Figures and Tables

**Figure 1 ijerph-18-01822-f001:**
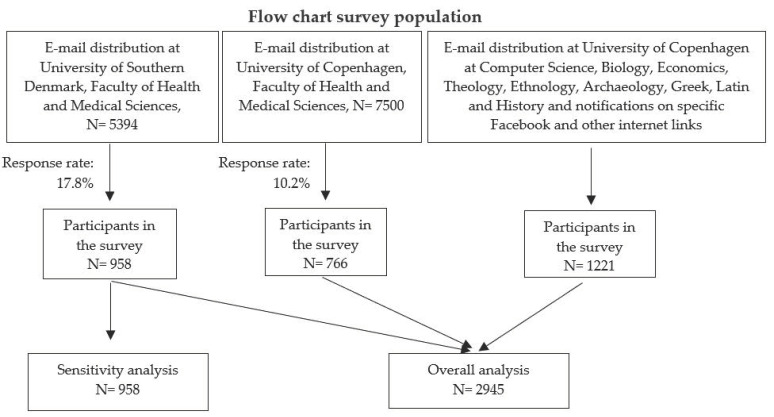
Flow chart survey population and analysis.

**Figure 2 ijerph-18-01822-f002:**
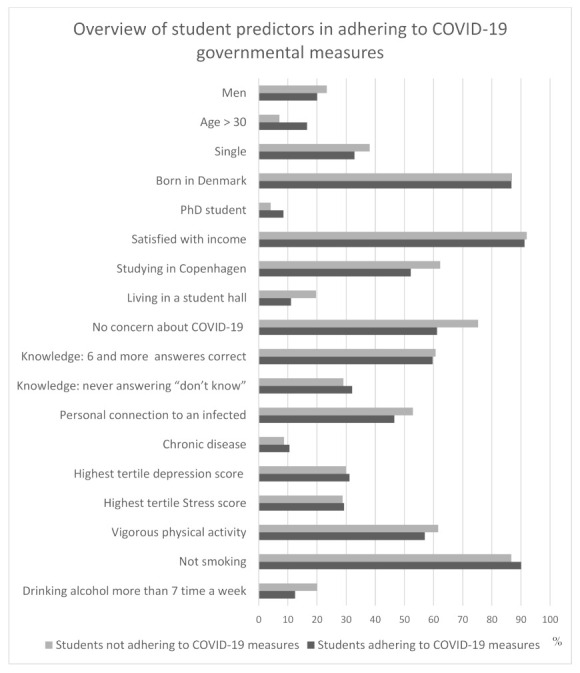
Prevalence of considered predictors to explain adherence to governmental COVID-19 measures for students who adhere and students who do not adhere to governmental COVID-19 measures.

**Table 1 ijerph-18-01822-t001:** Description of the study population, overall and in the sub-population adhering to COVID-19 measures that are implemented by the government.

	Overall		Students Adhering to COVID-19 Measures Implemented by the Government *		Students Not Adhering to COVID-19 Measures Implemented by the Government *	
	*n*	%	*n*	%	*n*	%
Overall *n*	2313	100.0	1578	100.0	735	100.0
Sex						
Men	489	21.1	317	20.1	172	23.4
Women	1809	78.2	1250	79.2	559	76.1
Other	15	0.7	11	0.7	<5	0.5
Age						
≤21	264	11.4	175	11.9	89	12.1
22–24	864	37.4	565	35.8	299	40.1
25–30	871	37.7	576	36.5	295	40.1
>30	314	13.6	262	16.6	52	7.1
Relationship						
Single	799	34.5	519	32.9	280	38.1
Born in						
Denmark	2008	86.8	1369	86.8	639	86.9
Study program						
Bachelor’s student	1082	46.8	708	44.9	374	50.9
Master’s student	1063	46.0	734	46.5	329	44.8
PhD student	164	7.1	134	8.5	30	4.1
Other **	-		-		-	
Income						
Satisfied with income	2116	91.5	1440	91.3	676	92.0
Place of study						
Area of Copenhagen	1282	55.4	824	52.2	458	62.3
Living situation						
With parents	131	5.7	93	5.9	38	5.2
Student hall	320	13.8	175	11.1	145	19.7
With others	1336	57.8	935	59.3	401	54.6
Alone	452	19.5	317	20.1	135	18.4
Other	74	3.2	58	3.7	16	2.2

* Adherence score above 7 points means adhering; 7 and below means not adhering; ** too small numbers to be presented.

**Table 2 ijerph-18-01822-t002:** Concern about COVID-19, health behavior and health-related variables in the whole student population, and in students with high adherence to COVID-19 measures that are implemented by the government.

	Overall		Students Adhering to COVID-19 Measures *		Students Not Adhering to COVID-19 Measures *	
	*n*	%	*n*	%	*n*	%
Overall *n*	2313	100.0	1578	100.0	735	100.0
Concern about being infected by COVID-19						
Not at all	1519	65.7	966	61.2	553	75.3
Medium	600	25.9	456	28.9	144	19.6
High	135	5.8	121	7.7	14	1.9
Already infected	59	2.5	35	2.2	24	3.3
Knowledge about COVID-19 **						
At least 6 out of 8 questions answered correctly	1388	60.0	942	59.7	446	60.7
Never answering “don’t know”	720	31.1	506	32.1	214	29.1
Personal connection to an infected person						
Yes	1125	48.6	736	46.6	389	52.9
Chronic disease						
present	231	10.0	167	10.6	64	8.7
Depression						
Low	1007	43.5	693	43.9	314	42.7
Middle	594	25.7	393	24.9	201	27.4
High	712	30.8	492	31.2	220	29.9
Stress **						
Low	860	37.2	607	38.5	253	34.4
Middle	747	32.3	491	31.1	256	34.8
High	674	29.1	462	29.3	212	28.8
Healthy behavior						
Vigorous physical activity	1353	58.5	900	57.0	453	61.6
Not smoking	2059	89.0	1422	90.1	637	86.7
*** Not drinking alcohol	513	22.2	380	24.1	133	18.1
Drinking alcohol 1–7 times a week	1453	62.8	999	63.3	454	61.8
Drinking alcohol more than 7 times a week	345	14.9	197	12.5	148	20.1

* Adherence score above 7 points means adhering; 7 and below means not adhering; ** 32 additional missing; *** 2 additional missing.

**Table 3 ijerph-18-01822-t003:** Simple and stepwise multiple linear regression to predict adherence to COVID-19 measures implemented by the government (*n* = 2278; adjusted r^2^ = 0.09).

	Simple Model		Multiple Model	
	Beta	95%CI	Beta	95%CI
Sex (ref. men)				
Women	**3.88**	**1.42; 6.33**	2.37	−0.09; 4.83
Age (ref. ≤21)				
22–24	−0.46	−3.82; 2.90	0.18	−3.26; 3.63
25–30	1.15	−2.20; 4.50	1.18	−2.69; 5.05
>30	**10.73**	**6.74; 14.72**	**7.40**	**2.68; 12.12**
Relationship (ref. in Partnership)				
Single	2.90	0.80; 5.01	--	
Born (ref. Outside Denmark)				
In Denmark	1.15	−1.81; 4.12	--	
Study program (ref. Bachelor’s student)				
Master’s student	**3.05**	**0.98; 5.12**	1.89	−0.50; 4.28
PhD student	**10.75**	**6.73; 14.77**	**5.20**	**0.58; 9.83**
Income (ref. Not satisfied with income)				
Satisfied with income	−0.23	−3.80; 3.37	--	
Place of study				
Area of Copenhagen	**−4.28**	**−6.30; −2.28**	--	
Living situation (ref. With parents)				
Student hall	**−11.51**	**−16.48; −6.55**	**−8.43**	**−13.44; −3.42**
With others	−2.49	−6.87; 1.89	−2.30	−6.84; 2.25
Alone	−2.96	−7.70; 1.79	−2.58	−7.40; 2.23
Concern about infection (ref. No concern)				
Medium	**8.15**	**5.88; 10.41**	**7.67**	**5.40; 9.95**
High	**19.97**	**15.74; 24.20**	**17.92**	**13.68; 22.17**
Infected in the past	−2.94	−9.18; 3.31	−1.52	−7.75; 4.70
Knowledge about COVID−19 ***				
At least 6 out of 8 questions answered correctly	−0.84	−2.91; 1.23	--	
Never answering “I don’t know”	1.53	−0.64; 3.70	--	
Personal relation to an infected person				
Yes	**3.17**	**1.16; 5.17**	1.94	−0.03; 3.91
Chronic disease (ref. No disease)				
Yes	**3.41**	**0.06; 6.75**	--	
Depression				
(Numerical)	0.13	−0.23; 0.51	**0.61**	**0.10; 1.11**
Stress				
(Numerical)	−0.16	−0.48; 0.14	**−058**	**−0.99; −0.17**
Health behavior				
(Ref: No activity) Vigorous physical activity	**−3.73**	**−5.76; −1.70**	−1.81	−3.81; 0.17
(Ref. No smoking) Smoking	**−3.47**	**−6.68; −0.26**	--	
(Ref: No drinking) Alcohol 1–7 times a week **	**−4.88**	**−7.34; −2.43**	−2.07	−4.53; 0.39
(Ref: No drinking) Alcohol > 7 times a week	**−10.76**	**−14.09; −7.42**	**−4.90**	**−8.32; −1.47**

* 32 missings. ** 2 missings; bold numbers mark significant results

**Table 4 ijerph-18-01822-t004:** Sensitivity analysis in repeating the stepwise multiple linear regression to predict adherence to COVID-19 measures implemented by the government only in health students from University of Southern Denmark (*n* = 850; adjusted r^2^ = 0.11).

	Multiple Model	
	Beta	95%CI
Sex (ref. men)		
Women	--	
Age (ref. ≤21)		
22–24	3.68	−1.82; 9.19
25–30	4.87	−0.71; 10.45
>30	**13.10**	**6.83; 19.37**
Relationship (ref. In partnership)		
Single	--	
Born in Denmark (ref. Outside Denmark)		
Denmark	−3.68	−7.79; 0.43
Study program (ref. Bachelor’s student)		
Master’s student	--	
PhD student	--	
Income (ref. Not satisfied with income)		
Satisfied with income	--	
Living situation (ref. With parents)		
Student hall	−6.93	−16.06; 2.20
With others	2.85	−5.21; 10.91
Alone	1.78	−6.43; 9.99
Concerned about infection (ref. Not concerned)		
Medium	**8.86**	**5.43; 12.29**
High	**19.96**	**13.35; 26.56**
Infected in the past	**16.12**	**1.53; 30.75**
Knowledge about COVID−19 ***		
At least 6 out of 8 questions answered correctly	--	
Never answering “I don’t know”	--	
Personal relation to an infected person		
Yes	--	
Chronic disease (ref. No disease)		
Yes	**−5.57**	**−10.83; −0.31**
Depression		
(Numerical)	**1.18**	**0.38; 1.98**
Stress		
(Numerical)	**−0.85**	**−1.51; −0.18**
Health behavior		
(Ref: No activity) Vigorous physical activity	−3.01	−6.20; 0.17
(Ref. No smoking) Smoking	--	
(Ref: No drinking) Alcohol 1–7 times a week **	--	
(Ref: No drinking) Alcohol > 7 times a week	--	

* 32 missings, ** 2 missings; bold numbers mark significant results

## Data Availability

The data presented in this study are available on request from the corresponding author. The data are not publicly available due to still ongoing international cooperation’s.

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
