# Peer review of "What Predicts Adherence to Governmental COVID-19 Measures among Danish Students?"

_ijerph, 2021, doi:10.3390/ijerph18041822_

Round 1
Reviewer 1 Report
Dear author(s):
Concerning the article entitled “What predicts adherence to governmental COVID-19 measures among Danish students?” (Manuscript ID: ijerph-1090535), which I have been asked to review, I hereby inform you of my decision:
"Reconsider after major revision".
The authors have done a rather original work, quite well structured and with some final conclusions that can be considered pretty interesting. However, I have found a series of "formal" defects in the article that does not recommend its definitive publication in its current state.
Synopsis of the review:
In general terms, this is such a good paper, performed with a fluent level of Academic English that, nevertheless, needs to be improved in the following fields:
- It is an excessively "short" work: it is not that the longer the work the better the final result should be, it is, under a minima point of view that they have not included some parts that should be basic. For example, I note that there is no section after the introduction that is specifically designed to describe the state of the literature (State of Art). Please include it. I know positively that you are analyzing a new subject, from a completely new study perspective, but I insist that you must include a minimal bibliographic review.
- In the final discussion, you should contextualize your results with the literature, that is, with other works of similar nature to yours that have been carried out in other countries.
- Line 303] "In conclusion, in Denmark, around 60% of the students were not concerned about COVID-19". Please reconstruct this sentence. In any case, it would be in line with your study, but to generalize to all of Denmark from such a small sample seems to me quite disproportionate.
- Lines 88-89] "Survey participation was voluntary and anonymous, and data were confidential and protected. I agree, but author(s) could at least indicate which Danish universities the work was based on. All of them? Some? Please specify.
- Survey design. Could author(s) please include all the questions asked to the students in table form as a final appendix? If they are very extensive, please summarize them.
- Figures. As I mentioned before, the fact of including figures or not, is not indicative of a better or worse job, but in your case, I consider that your article would gain a lot. I recommend that author(s) include several pie charts directly related to table 1 (for instance). In this way the reading of the article will be more "friendly" for a future reader.
- Additional questions. In Denmark there is a huge population group with an immigrant background and a large number of nationalities and cultures coexist. Probably among the university students who answered your questionnaire, it has been the case that. Please indicate in your work whether or not this population group has been taken into account.
I do repeat again, you have produced a great paper. I encourage you to make the suggestions that were highlighted: For my part, I would be delighted if this work were published, once the authors(s) have made the changes that I have pointed out.
With my best wishes to your family and friends in 2021,
The reviewer.
Author Response
We thank the reviewers for their constructive comments. We appreciated their work. Please find below our responses and amendments in blue to the reviewers’ comments.
When revising the manuscript, we highlighted the changes by using the track changes mode in MS Word.
We hope that these are to the satisfaction of the reviewers and that the manuscript can now be accepted for publication.
Sincerely,
Gabriele Berg-Beckhoff
Reviewer 1:
- It is an excessively "short" work: it is not that the longer the work the better the final result should be, it is, under a minima point of view that they have not included some parts that should be basic. For example, I note that there is no section after the introduction that is specifically designed to describe the state of the literature (State of Art). Please include it. I know positively that you are analyzing a new subject, from a completely new study perspective, but I insist that you must include a minimal bibliographic review.
Thank you, we appreciate your comment; we added an additional chapter about the state of art in this research. You can find additions in the first and second chapter of the introduction.
- In the final discussion, you should contextualize your results with the literature, that is, with other works of similar nature to yours that have been carried out in other countries.
Thank you for this comment. We ordered the discussion to fit the suggested predictors given in the introduction and we added additional literature.
- Line 303] "In conclusion, in Denmark, around 60% of the students were not concerned about COVID-19". Please reconstruct this sentence. In any case, it would be in line with your study, but to generalize to all of Denmark from such a small sample seems to me quite disproportionate.
Thank you for this careful reading, that is absolutely correct we reformulated the sentence to “In conclusion, in our survey, around 60% of the students were not concerned about COVID-19".
- Lines 88-89] "Survey participation was voluntary and anonymous, and data were confidential and protected. I agree, but author(s) could at least indicate which Danish universities the work was based on. All of them? Some? Please specify.
The universities are mentioned while explaining the procedures. We like to keep this there. We clarified that beside the e-mail invitation to University of Southern Denmark and University of Copenhagen additional students participated following promotion in facebook pages allowed also entries from other universities.
- Survey design. Could author(s) please include all the questions asked to the students in table form as a final appendix? If they are very extensive, please summarize them.
A very good suggestion, as an English translation of the questionnaire is available. We added a document containing questions/answers of all questions that are used for the present analysis. This table is now added as supplementary file.
- Figures. As I mentioned before, the fact of including figures or not, is not indicative of a better or worse job, but in your case, I consider that your article would gain a lot. I recommend that author(s) include several pie charts directly related to table 1 (for instance). In this way the reading of the article will be more "friendly" for a future reader.
We added one diagram on descriptive information. Thank you for the suggestion (see figure 2).
- Additional questions. In Denmark there is a huge population group with an immigrant background and a large number of nationalities and cultures coexist. Probably among the university students who answered your questionnaire, it has been the case that. Please indicate in your work whether or not this population group has been taken into account.
Thank you very much, that is important to note, we have international students that also participated. The survey questionnaire was also provided in English to take care that all students can answer. This is added into the document (1) in the method part as well as in the (2) result section the percentage of students born outside Denmark is named specifically.
Reviewer 2 Report
This manuscript does an excellent job demonstrating how educated people respond to the government measures about the covid-19. This paper shows interesting data, but it shows a clear bias. In other words, most college students and workers had to adhere to the school’s policies, or they will be banned or sent to quarantine by the school officials.
Overall, this article about the covid-19 response in highly educated settings is an addition to the covid-19 research, especially after pandemic response and reopening. Title and abstract. The title is appropriate for the work and the abstract is concise and accurate.
Case report.
• It will be great if the author summaries the Danish government guidelines for reopening and how those were implemented.
• How does the author control multiple responses in the survey? What variables were used?
• Please clarify if the survey/study was made with a major group of international students. In other words, most of the participants or volunteers were international students (line 88).
• How was the flowchart of the survey developed for this study?
• Since the author mention that this is a part of another study: is this already publish data and was analyzed only a section, in this study the Danish case? • In table 1. Where are people who don’t adhere to the policies? It will be a good comparison.
• Why does table 1 only shows relationship: single? No married people at all? Data shows age >30 = 314. Conclusion. What predicts adherence to governmental COVID-19 measures 2 among Danish students? The author mentioned older age, but the survey shows >30. Most publications show the same conclusion, and it is well known that age affects the concerns about covid-19.
Also, the author mention that “Main facilitators for following the recommendation were older age, concern about COVID-19, and depression (line 305)” and “Another interesting finding was the fact that depression and mental stress were not 267 associated with adherence to COVID-19 measures in bivariate analyses (LINE 267).” It is a little contradiction in the conclusion.
Author Response
We thank the reviewers for their constructive comments. We appreciated their work. Please find below our responses and amendments in blue to the reviewers’ comments.
When revising the manuscript, we highlighted the changes by using the track changes mode in MS Word.
We hope that these are to the satisfaction of the reviewers and that the manuscript can now be accepted for publication.
Sincerely,
Gabriele Berg-Beckhoff
Reviewer 2:
This manuscript does an excellent job demonstrating how educated people respond to the government measures about the covid-19. This paper shows interesting data, but it shows a clear bias. In other words, most college students and workers had to adhere to the school’s policies, or they will be banned or sent to quarantine by the school officials.
Dear reviewer, thank you for your nice words, and your important comment. Your mentioned bias is now mentioned under limitations. We added the following section:
Finally, we cannot exclude that most college students had to adhere to the Governmental policies, as e.g. the universities were closed, and it was impossible to enter the buildings. However, other recommendation like distance, hand hygiene and meeting restriction still needed students’ own decision. Furthermore, the survey took place in the reopening phase, in which the support for the COVID-19 measures started to erode in society. The controversial public discussion facilitates free opinion formation of the students.
Overall, this article about the covid-19 response in highly educated settings is an addition to the covid-19 research, especially after pandemic response and reopening. Title and abstract. The title is appropriate for the work and the abstract is concise and accurate.
Case report.
- It will be great if the author summaries the Danish government guidelines for reopening and how those were implemented.
Thank you for this remark we added information. However, there were many activities and guidelines that we decided not to add them all. Therefore, we added the link to the WHO COVID-19 health monitor system, where all guidelines and activities are careful presented for several countries including Denmark.
- How does the author control multiple responses in the survey? What variables were used?
The questionnaire is now attached as supplementary file. We did not have any problem with multiple answers. Only in one question about chronic disease, multiple answers were allowed. However, by building the dichotomous category any disease (it did not matter how many) and no disease, multiple answers were solved.
- Please clarify if the survey/study was made with a major group of international students. In other words, most of the participants or volunteers were international students (line 88).
Thank you very much, that is important to note, we have international students that also participated. The survey questionnaire was also provided in English to take care that all students can answer. This is added into the document (1) in the method part as well as in the (2) result section the percentage of students born outside Denmark is named specifically.
- How was the flowchart of the survey developed for this study?
A good suggestion, a flowchart was added as figure 1.
- Since the author mention that this is a part of another study: is this already publish data and was analyzed only a section, in this study the Danish case? • In table 1. Where are people who don’t adhere to the policies? It will be a good comparison.
Good suggestion number are added to table 1 (and also to table 2)
- Why does table 1 only shows relationship: single? No married people at all? Data shows age >30 = 314. Conclusion. What predicts adherence to governmental COVID-19 measures 2 among Danish students? The author mentioned older age, but the survey shows >30. Most publications show the same conclusion, and it is well known that age affects the concerns about covid-19.
The question was adapted to the situation in students; it was not asked if they were married. The question was: Are you currently in a steady relationship? And the answers were: (1) no I am single (2) yes, (3) it’s complicated. For the analysis, we summed up the 2 and 3. Therefore, we do not have information about married students. All used questions and the corresponding answers are now added in an additional supplemental document for further information.
Yes, age is an important factor explaining adherence to governmental recommendation, which can be seen in all multiple analysis. Due to the small number of older students, we summed up all students above 30 in one group.
Also, the author mention that “Main facilitators for following the recommendation were older age, concern about COVID-19, and depression (line 305)” and “Another interesting finding was the fact that depression and mental stress were not 267 associated with adherence to COVID-19 measures in bivariate analyses (LINE 267).” It is a little contradiction in the conclusion.
Thanks for your careful reading that looks contradictive. We clarify this by adding in the first discussion paragraph about the result summary, from where the results are. In the multiple regression model (where the model was adjusted for all confounding variables) depression is a facilitator and stress a barrier, but in the crude or bivariate model (the models without adjustment) no association is found. Additionally, we changed the order of presenting the results. However, the main message in this paragraph is that results are contradictive. This contradiction is not only in our analysis, it is also given in other publications. Further research is necessary.
Round 2
Reviewer 1 Report
Reviewer report
Dear Author(s):
In accordance with the review of the article "What predicts adherence to governmental COVID-19 measures among Danish students?" (ijerph-1090535), which has been assigned to my person, I communicate below what my decision was:
Accept in present form.
Synopsis of the review
Indeed, the authors have followed the changes, suggestions, and proposals for improvement requested by this reviewer. Furthermore, I consider that the final version of the article, as a result of my review and the other referees', has represented a significant improvement on the initial version, so I would be deeply flattered if this work is finally accepted.
I would just like to make a few minor clarifications, a few small nuances that I would like you to take into account:
I find the idea of including the survey design as a supplementary file extraordinary, but I still consider that it would have been much better to include it as an appendix at the end of the work, which would not have meant a noticeable effort on the part of the authors. Likewise, I still do not exactly see the names of the Danish universities whose students responded to the survey. This is probably an error on my part, but if not, I think it would be important to mention them.
In any case, I must again congratulate the five authors for their work.
Best regards,
The reviewer.